# IL-1 Family Cytokines in Inflammatory Dermatoses: Pathogenetic Role and Potential Therapeutic Implications

**DOI:** 10.3390/ijms23169479

**Published:** 2022-08-22

**Authors:** Helena Iznardo, Luís Puig

**Affiliations:** 1Dermatology Department, Hospital de la Santa Creu i Sant Pau, Universitat Autònoma de Barcelona, 08041 Barcelona, Spain; 2Institut d’Investigació Biomèdica Sant Pau (IIB SANT PAU), 08041 Barcelona, Spain

**Keywords:** psoriasis, pustular psoriasis, hidradenitis suppurativa, atopic dermatitis, imsidolimab, spesolimab, bermekimab, etokimab, IL-36, IL-36R, IL-1, IL-33, pathogenesis

## Abstract

The interleukin-1 (IL-1) family is involved in the correct functioning and regulation of the innate immune system, linking innate and adaptative immune responses. This complex family is composed by several cytokines, receptors, and co-receptors, all working in a balanced way to maintain homeostasis. Dysregulation of these processes results in tissue inflammation and is involved in the pathogenesis of common inflammatory dermatoses such as psoriasis, hidradenitis suppurativa, and atopic dermatitis. Therefore, therapeutic targeting of IL-1 pathways has been studied, and several monoclonal antibodies are currently being assessed in clinical trials. So far, promising results have been obtained with anti-IL-36R spesolimab and imsidolimab in pustular psoriasis, and their efficacy is being tested in other conditions.

## 1. Introduction

IL-1 family members are central players of the immune system. They are especially involved in the regulation of innate immune responses, maintaining endogenous hemostasis, and linking innate and adaptive responses. Several cytokines, receptors, and accessory proteins constitute this complex family; their activation and expression are balanced by different regulatory mechanisms, and their disturbance results in pathologic inflammatory responses. Disruption of IL-1-related pathways is involved in several inflammatory dermatoses such as psoriasis, hidradenitis suppurativa (HS), atopic dermatitis (AD), as well as several neutrophilic dermatoses. In this comprehensive narrative review, we will discuss the biological particularities of IL-1 family members, their involvement in cutaneous inflammatory diseases, and the current therapeutic strategies targeting this complex pathway. 

## 2. IL-1 Family Cytokines, Receptors and Co-Receptors

The IL-1 family of cytokines is composed of 11 cytokine members, with seven agonists (IL-1α, IL-1β, IL-18, IL-33, IL-36α, IL-36β, and IL-36γ) and four antagonists (IL-1 receptor antagonist (Ra), IL-36Ra, IL-37, and IL-38) [1]. According to their structural and functional characteristics, these cytokines are further classified into four subfamilies (IL-1, IL-18, IL-33, and IL-36), each one having a cognate receptor (IL-1R1, IL-18Rα, IL-33R (suppression of tumorigenicity 2 or ST2), and IL-36R, respectively). Furthermore, IL-1RAcP is an accessory protein shared by all these cytokines, with the exception of IL-18 (IL-18RAcP or IL-18Rβ chain) (Table 1) [2].

To produce their pro-inflammatory functions, IL-1 cytokines form complexes with their respective receptor and a co-receptor (Figure 1). All of them except IL-1Ra are synthesized as precursors and require N-terminal processing in order to acquire their full function [3,4]. They can be activated both extracellularly by proteolytic cleavage and intracellularly via inflammasome-mediated cleavage [5]. Binding of cleaved IL-1α/β to the extracellular domain of IL-1R1 leads to recruitment of IL-1RAcP, resulting in the initiation of a signaling cascade with the recruitment of the myeloid differentiation primary response 88 (MyD88) accessory protein and Interleukin 1 receptor-associated kinases (IRAKs). This in turn results in activation of the nuclear factor κB (NF-κB) and mitogen-activated protein kinases (MAPKs), ultimately resulting in pro-inflammatory gene expression [6]. Furthermore, IL-1 signaling also induces activation of defense mechanisms (antigen recognition, phagocytosis, degranulation, and nitric oxide production) and activates lymphocyte functions implicated in adaptive immunity, thus acting as a link between innate and adaptive immune responses [7]. The other IL-1 family cytokines IL-33, IL-18, and IL-36α/β/γ form similar ternary complexes with their respective receptors and co-receptors and also act through Myd88 to induce pro-inflammatory gene expression.

Regulatory mechanisms are necessary to maintain homeostasis; they include decoy receptors, receptor antagonists, and anti-inflammatory cytokines (Figure 1). IL-1R2 is a cytoplasmic soluble receptor without a functional TIR domain that binds to IL-1α/β precursors, preventing their processing and secretion. Under proinflammatory conditions, IL-1R2 is cleaved by an inflammasome-dependent mechanism [8,9]. Likewise, the soluble ST2 receptor (sST2) and the soluble protein IL-18 binding protein (IL-18BP) bind to IL-33 and IL-18, respectively, neutralizing their activities [10,11]. Furthermore, receptor antagonists IL-1Ra, IL-36Ra, and IL-38 compete with IL-1α/β and IL-36α/β/γ [4]. Lastly, IL-37 binding to IL-18Rα leads to recruitment of the IL-1R8 co-receptor (also called single immunoglobulin IL-1R-related molecule (SIGIRR), with activation of the inhibitory STAT3 signaling pathway [12].

### 2.1. IL-1 Subfamily

IL-1α and IL-β are both pro-inflammatory cytokines with some distinctive characteristics. IL-1α is constitutively expressed in hematopoietic immune cells and other cell types such as intestinal epithelial cells and cutaneous keratinocytes (KCs) [13]. Although non-processed full-length IL-1α has some functional activity, cleavage by calpain and extracellular proteases such as neutrophil elastase, granzyme B, and mast cell chymase enhances its pro-inflammatory activity [14,15]. Its major role is played locally, since IL-1α is mostly found bound to membranes. It is also expressed intracellularly in the cytosol—acting as an alarmin upon release from necrotic cell death—and in the nucleus—activating transcription of pro-inflammatory genes or tissue homeostasis and repair genes [16]. In addition, IL-1α expression can be induced by proinflammatory stimuli, leading to IL-1R1 binding and pro-inflammatory gene expression targeting type 1 or type 17 immune responses. In turn, this produces recruitment and activation of T cells, dendritic cells (DCs), neutrophils, and monocytes/macrophages that will release further pro-inflammatory cytokines and chemokines, leading to an autoinflammatory amplification loop [17]. On the contrary, IL-1β is the primary circulating form, and its expression is inducible only in monocytes, macrophages, and DCs [17]. Full-length IL-1β precursor protein (pro-IL-1β) is stored in the cytoplasm and is cleaved by caspase-1 to its active form in response to activation of pattern recognition receptors (PPR) by pathogen-associated molecular patterns (PAMPs) or damage-associated molecular patterns (DAMPs) in an inflammasome-dependent process [18]. In addition, pro-IL-1β can also be activated in the extracellular space by neutrophils and mast cells-derived proteases or by microbial proteases [19]. The antagonist IL-1Ra exerts its anti-inflammatory properties by binding to the IL-1R1 receptor and competing with IL-1α and IL-β [20].

### 2.2. IL-18 Subfamily

IL-18 is also a pro-inflammatory cytokine and is constitutively expressed in its inactive form in several cell types, mainly KCs, epithelial, and endothelial cells [21]. Its activation can be produced intracellularly by caspase-1-mediated cleavage or extracellularly by neutrophil or cytotoxic cell-derived proteases [21,22]. IL-18 pro-inflammatory activity is mainly mediated through IFNγ, but it can induce both Th1 and Th2 cellular responses [23]. In combination with IL-12 and IL-15, IL-18 stimulates Th1 cells and induces NK cell effector function and IFNγ expression [24,25]; in the absence of these cytokines, IL-18 induces a Th2 response with mast cell and basophil activation, ultimately ending in IL-4 and IL-13 production [26]. Moreover, when combined with IL-23, IL-18 activates Th17 cells and induces IL-17 production [27]. IL-18-BP regulates IL-18 pro-inflammatory activity by binding and sequestering the cytokine. IL-18-BP expression is induced by IFNγ, thus creating a negative feedback mechanism and decreasing inflammation [11]. 

### 2.3. IL-33 Subfamily

IL-33 is also constitutively expressed in many organs, mainly by fibroblasts, endothelial cells, and epithelial cells; its expression can also be induced in mast cells and DCs in the context of inflammation. Similar to IL-1α, IL-33 requires cleavage to increase its activity and has a dual role: intracellular gene expression regulating homeostasis and extracellular recruitment and activation of immune cells upon cell necrosis and inflammation (alarmin function) [28,29]. Th2 cells, mast cells, and eosinophils express the IL-33 receptor, ST2, or IL1-1RL1 [28]. IL-33 is a promoter of Th2 immunity and allergic responses, inducing production of IL-4, IL-5, and IL-13, polarization of macrophages and degranulation of mast cells, basophils, and eosinophils with cytokine and chemokine release [30]. Finally, IL-33 also acts on T-reg cells, DCs, and NK cells [31].

### 2.4. IL-36 Subfamily 

The IL-36 subfamily is a key regulator of the innate immune system and includes three agonists with pro-inflammatory activity (IL-36α, IL-36β, and IL-36γ) and two antagonists (IL-36RN or IL-36Ra and IL-38) [32]. They are normally expressed in epithelial and immune cells; after binding to receptor complex IL-36R, agonists induce activation of nuclear factor-kB (NF-kB) and mitogen-activated protein kinases, leading to T-cell proliferation, expression of pro-inflammatory cytokines, chemokines, and co-stimulatory molecules by DCs and Th1 lymphocytes, as well as autocrine KCs signaling. The resulting pro-inflammatory milieu is composed of IL-1β, IL-12, IL-23, IL-6, TNF-α, CCL1, CXCL1, CXCL2, CXCL8, and GM-CSF, among others [32]. IL-36α and IL-36γ are mainly produced by KCs but also by dermal fibroblasts, endothelial cells, macrophages, LCs, and DCs. As opposed to other IL-1 family cytokines, IL-36 cytokines are also produced as precursors but do not contain a caspase cleavage site. Following secretion, they are activated by neutrophil-derived proteases present at neutrophil extracellular traps (NETs)—such as elastase, cathepsin G, and proteinase 3—and by cathepsin S, produced by KCs and fibroblasts [33,34,35]. In addition, KCs secrete the protease inhibitors alpha-1-antitrypsin and alpha-1-antichymotrypsin (encoded by SERPINA1 and SERPINA3 genes), which inhibit processing of IL-36 cytokines by neutrophil proteases and thus regulate the inflammatory loop [36].

### 2.5. IL-37 and IL-38: Antagonist Cytokines

IL-37 acts as an anti-inflammatory cytokine and is constitutively expressed mainly in KCs, but can be induced in monocytes/macrophages, T cells, and B cells. It has a dual action, depending on extracellular or intracellular signaling. Extracellularly, IL-37 binds IL-18Rα and recruits IL-1R8 to form the IL-37/IL-1R8/IL-18Rα complex, restricting IL-18R-dependent inflammation and inhibiting innate immunity [12]. In the cytosol, IL-37 cleaved by caspase-1 translocates to the nucleus to bind Smad transcription factors, ultimately decreasing pro-inflammatory cytokine production. The precursor exhibits activity, but cleaved IL-37 binds more effectively to its receptor [37].

IL-38 also acts as an anti-inflammatory cytokine and is expressed in skin and various immune cells, such as B cells. It specifically binds to IL-36R and inhibits human mononuclear cells stimulated with IL-36 in vitro [38]. IL-38 expression is inhibited by IL-17, IL-22, and IL-36γ [4]. Furthermore, IL-38 is able to suppress the production of IL-17A by γδ T-cells through IL-1RAcP antagonism [39].

## 3. Involvement of IL-1 Family Cytokines in Inflammatory Dermatoses

### 3.1. Psoriasis

Psoriasis is a chronic inflammatory multisystem disorder with an immunogenetic basis and a pathogenesis characterized by unbalanced interactions between the innate and the adaptive immune systems. The clinical presentation of psoriasis is morphologically and topographically heterogeneous; the main variants are plaque psoriasis—also known as psoriasis vulgaris—and pustular psoriasis (PP). Although there is frequent clinical overlap, their clinical, genetic, and pathogenetic distinguishing features suggest that they are in fact distinct entities [36,40,41]. In the pathogenesis of plaque psoriasis, there is a predominant involvement of the adaptive immune system, with a critical role of the IL-17/23 axis. Th1 and Th17 lymphocytes release TNF, IL-17, and IL-22, promoting proliferation of KCs and secretion of proinflammatory chemokines that lead to a self-amplification loop [42]. In contrast, autoinflammation and innate immune system activation are the main drivers of PP, with neutrophil infiltration of the epidermis and IL-36 and IL-1 cytokines acting as key pathogenic orchestrators of the disease [43].

The effects of several members of the IL-1 cytokine family have been studied in both animal and human models of psoriasis. Murine models have shown that IL-1α is able to initiate spontaneous cutaneous inflammation with histological similarities to psoriasis lesions and to stimulate KCs to induce potent proinflammatory responses [44,45]. IL-1β also stimulates epidermal KCs to secrete inflammatory chemokines and has proven to be critical in inducing Th17 and γδT17 cell differentiation and effector functions [45]. Moreover, IL-1 cytokines can stimulate mast cells to produce IL-6, tumor necrosis factor (TNF), and IL-33 without degranulation [38]. Overexpression of IL-18 has been detected in both skin lesions and peripheral blood of psoriasis patients, with IL-18 serum levels directly correlating with psoriasis severity [46]. In a mouse model, IL-18 also induced prominent inflammation, with increased expression of IFNγ and enhancement of psoriasis-like epidermal hyperplasia [47]. IL-33 also appears to be involved in the pathogenesis of psoriasis, participating in the crosstalk between innate and adaptive responses. In KCs, IL-33 works in an autocrine loop; it is produced by KCs following psoriatic inflammatory stimuli and induces KCs transcription of pro-inflammatory genes (CCL2, CXCL1, CXCL2, CXCL15, and vascular endothelial growth factor) [48]. Increased IL-33 serum levels have been found in patients with psoriasis, and they were significantly reduced after treatment with TNF inhibitors [30,49].

The discovery of deficiency of IL-36 receptor antagonist (DITRA), a recessive autoinflammatory syndrome due to loss-of-function mutations in the gene IL36RN (encoding IL-36Ra), highlighted the role of IL-36 cytokines in PP. In DITRA, unopposed signaling of IL-36 cytokines results in a severe PP phenotype with repeated flares of multiple pustules and fever, leukocytosis, and elevated serum levels of C-reactive protein [50]. Moreover, in murine and human models, KCs surrounding neutrophilic pustules have shown elevated expression of IL-36 [36]. However, IL-36 cytokines are also involved in plaque psoriasis, with increased expression of IL-36α, IL-36β, and IL-36γ in skin and serum of psoriasis patients, and a positive correlation between disease severity and cytokine levels [18,51].

Regarding inhibitory cytokines, several studies have found decreased expression of IL-38 in both skin and blood of psoriasis patients, whereas its expression was increased in normal skin and upregulated in lesional psoriasis following treatment with anti-IL-17A biologic agents [52,53,54]. Lastly, IL-37 is able to inhibit IL-1β, IL-6, TNF, and chemokines such as CCL2, all involved in psoriasis [55]. Therefore, IL-38 and IL-37 are probably also involved in the pathogenic pathways of psoriasis.

### 3.2. Atopic Dermatitis

AD is a heterogeneous, chronic inflammatory dermatosis with a relapsing and remitting course. Its complex pathogenesis involves skin barrier dysfunction, transepidermal water loss, immune system abnormalities with increased predisposition to infection, microbial dysbiosis, and neurosensory abnormalities [56,57]. Activation of type 2 immune response is of paramount importance, with variable participation of Th1 and Th17/IL-23 pathways in some phenotypes [58,59]. Most cases of AD can be classified as extrinsic AD, with elevated IgE levels, filaggrin mutations, and a deficient skin barrier. On the contrary, intrinsic AD is associated with increased activation of Th1 and Th17 cells [60,61]. Increased Th2 cell responses with overexpression of IL-4, IL-5, IL-13, and IL-31 are consistently found in both intrinsic and extrinsic AD patients [62].

Regarding IL-1 family cytokines, increased expression of IL-36α and IL-36γ in both serum and skin has been found in patients with AD in comparison with healthy controls [63]. In mouse models of AD-like skin inflammation induced by epicutaneous exposure to *Staphylococcus aureus*, treatment with anti-IL-36R-blocking antibodies suppressed release of IL-36α by KCs, IL-4 triggered B cell IgE class-switching, plasma cell differentiation, and increased serum IgE levels [63]. DITRA patients (increased IL-36 signaling) also have elevated serum IgE levels [64]. IL-33 levels have been found to be elevated in the serum of AD patients, and transgenic mice with enhanced expression of the IL-33 gene show an AD-like phenotype [65]. Moreover, IL-33 is able to induce Th2 cell differentiation and to promote the expression of IL-31 by Th2 cells [66].

Furthermore, progression from acute to chronic AD is associated with enhanced dysregulation of Th1, Th2, and Th17 responses, along with increased IL-36 expression, suggesting that IL-36 might play a role in lesion progression [67]. Finally, upregulation of IL-36α was found in both intrinsic AD and psoriasis patients, as opposed to healthy controls [68]. These data point towards a predominant role of IL-36 in AD subtypes with enhanced Th1 and Th17 inflammation [43]. Finally, treatment of KCs with IL-36 led to decreased expression of filaggrin, suggesting a possible involvement of IL-36 cytokines in barrier deficiencies in AD [69].

### 3.3. Hidradenitis Suppurativa

HS is another heterogeneous, chronic, and relapsing skin disease characterized by the inflammation of hair follicles in apocrine gland-bearing areas [70]. Its pathogenesis is complex, implying immune dysregulation, environmental factors, and genetic predisposition. A simplistic description of the immunological factors involved in HS include release of DAMPs and PAMPs, activation of Th1 (IL-12 and TNFα) and Th17 (IL-23 and IL-17) pathways, activation of macrophages through TLR with induction of TNFα, and activation of the inflammasome with subsequent IL-1β production [70,71,72].

Hyperactivation of the IL-1 pathways is of paramount importance in the pathogenesis of HS. IL-1β produced by KCs and macrophages/monocytes induces a strong production of chemokines, contributing to neutrophile infiltration and purulent discharge. In addition, IL-1β enhances the secretion of matrix-degrading enzymes, ultimately resulting in tissue destruction [73]. Furthermore, a study found that IL-1β mRNA and protein levels were strongly elevated in lesional HS skin compared with healthy skin, as opposed to IL-1RA levels; this results in an increased IL-1 β/IL-Ra ratio. Overexpression of these cytokines is more pronounced in HS than in lesional psoriatic skin [73]. Expression of granulocyte colony-stimulating factor (G-CSF), a key regulator of neutrophil survival and function, is increased in both skin lesions and blood of HS patients, and blood levels were positively correlated with severity of HS [74]. In cellular models, G-CSF expression in fibroblasts and KCs is induced by IL-1β, IL-17, and IL-36 [74].

Significant upregulation of IL-36 family cytokines has also been found in lesional HS skin (IL-36α, IL-36β, IL-36γ, and IL-36Ra) and in perilesional HS skin (IL-36β and IL-36Ra) compared to skin of healthy controls [75,76]. However, when comparing the expression of IL-36 cytokines in HS lesional skin vs. psoriasis lesional skin, higher levels are found in the latter [77]. Finally, increased serum levels of IL-36α, IL-36β, and IL-36γ were detected in HS patients, and they were also higher in smoking patients [78]. In fact, increased levels of IL-36 cytokines were found to be associated with increased risk of HS, even after controlling for HS comorbidities (smoking, obesity, and metabolic syndrome) [78].

### 3.4. Other Dermatoses

Allergic contact dermatitis (ACD) is an inflammatory dermatosis caused by skin contact with allergens. Increased levels of IL-36 cytokines have been found in lesional skin of ACD patients, compared to healthy skin from the same patients and normal controls. A strong upregulation of IL-36 cytokine expression was observed after ex vivo allergen challenge of uninvolved skin from ACD patients (in skin cultures). Injection of recombinant IL-36Ra suppressed the expression of these cytokines [79].

Neutrophilic dermatoses are characterized by sterile cutaneous inflammation with neutrophilic infiltrates. Examples include Sweet syndrome (SS), pyoderma gangrenosum (PG), and acute generalized exanthematous pustulosis (AGEP). SS is characterized by sudden development of erythematous, edematous, and painful cutaneous lesions accompanied by fever and leukocytosis. PG presents with painful skin ulcers with undermined borders and peripheral erythema and can be idiopathic or appear in the context of intestinal bowel disease. AGEP is a severe cutaneous adverse drug reaction with clinical and histological features shared with PP [40]. Although the pathogenesis of neutrophilic dermatoses is not completely understood, dysregulation of the innate immune system appears to be one of the main mechanisms involved [80]. Increased levels of IL-1b gene expression have been found in SS and PG [81,82]. Furthermore, expression of IL-36γ has been found to be increased in the epidermis during AGEP; culprit drugs can stimulate KCs to secrete IL-36γ, with subsequent IL-8 production by macrophages and T cells and chemotaxis of neutrophils in skin lesions of AGEP [83].

## 4. Therapeutic Targeting of IL-1 Family Cytokines

### 4.1. IL-1

Several IL-1 blocking agents have been developed to treat different diseases; the corresponding indications have been approved in some cases, and there are several ongoing clinical trials, with variable results.

Anakinra is a recombinant IL-1 receptor antagonist that competitively inhibits the interaction of both IL-1α and IL-1β with their receptor. It is currently approved for the treatment of rheumatoid arthritis and cryopyrin-associated periodic syndromes. Anakinra has shown efficacy in deficiency of interleukin-1 receptor antagonist (DIRA) and variable responses have been obtained in PP patients [84,85]. The results of the APRICOT trial—a double-blind, multicenter, 8-week placebo-controlled trial to determine the efficacy of anakinra for the treatment of adults with palmoplantar pustulosis (PPP)—have shown futility of anakinra in 64 patients [86]. However, poor adherence due to the daily injection schedule, short duration of the intervention, and a different pathogenesis in PPP could be confounding factors influencing the poor response to anakinra in this trial [86]. Anakinra has been used in HS with controversial results. Case reports and small series have shown variable degrees of efficacy, with an open-label study of five patients finding reductions in disease activity [87]. A double-blind, randomized, placebo-controlled trial had promising results (78% of treated patients achieved at least a 50% reduction in the inflammatory lesions vs. 30% in the placebo group) [88]. However, long-term results from a more recent small study showed eventual relapse and class switching from anakinra to other therapeutic strategies [89]. Anakinra was assessed in a phase II clinical trial in inflammatory pustular skin diseases including Sneddon–Wilkinson disease, acrodermatitis continua of Hallopeau, PP, and PPP. Fifty percent of participants achieved 50% disease improvement by the end of week 12 [90].

Bermekimab, an inhibitor of IL1α, has yielded good results in phase II open-label studies in HS patients—even after failure to anti-TNF therapy—with significant reductions in inflammatory lesions and no significant drug-related adverse events [91,92,93]. Currently, there is one ongoing trial with bermekimab (NCT04988308) in HS [94]. Furthermore, it was used in 11 patients with acne vulgaris, with rapid improvements and no severe adverse events [95]. In an open-label trial, bermekimab also showed encouraging results in psoriasis [96]. Two phase 2 studies of bermekimab for the treatment of adult patients with moderate-to-severe AD were terminated due to lack of efficacy [97,98].

Canakinumab is a human monoclonal antibody targeting IL-1β and has shown contradictory efficacy results in case reports in HS [99,100]. In psoriasis, canakinumab has been reported to be efficacious in a patient with a severe form of PP [101], but not in two patients with severe PPP [102]. In an open-label prospective study of steroid-refractory PG, canakinumab provided complete remission in 3/5 patients [82]. 

Gevokizumab is a novel IL-β inhibitor that has shown efficacy in PP patients without prior history of plaque psoriasis [91].

### 4.2. IL-18

Tadekinig alfa is a human recombinant IL-18-binding protein that is being investigated in a phase II open-label clinical trial on 23 patients with adult onset Still disease, a systemic inflammatory auto-inflammatory disorder characterized by fever, arthralgia, cutaneous eruption, and leukocytosis. A favorable safety profile with a clinical and biochemical response rate of 50% has been reported [103].

### 4.3. IL-33

Etokimab (ANB020) is an anti-IL-33 humanized monoclonal antibody being currently assessed in AD patients, with an ongoing phase IIb trial. A Phase IIa proof-of-concept clinical trial showed improvement in clinical score and reduced skin neutrophil infiltration as well as peripheral eosinophil counts after a single systemic administration of etokimab [104]. Results from a phase II study in adults with chronic rhinosinusitis with nasal polyps have already been posted [105]. 

REGN3500 is another monoclonal antibody targeting IL-33 that was under investigation in patients with AD, but development was terminated due to lack of efficacy [106,107]. 

PF-06817024 is an anti-IL-33 monoclonal antibody; a phase I placebo-controlled trial in healthy subjects, patients with chronic nasal sinusitis, and patients with AD has been recently completed, with no evidence of serious adverse events [108]. 

A monoclonal antibody targeting IL-33R (CNTO7160) is being investigated in AD. In a phase I clinical trial in patients with asthma, AD, and healthy individuals, effective inhibition of the IL-33R signaling pathway was observed, although this did not translate into significant clinical improvement [109]. 

### 4.4. IL-36

Monoclonal antibodies targeting IL-36R are currently being assessed in several diseases, with promising results in PP (Table 2).

Spesolimab (BI655130, Boehringer Ingelheim, 900 Ridgebury Road P.O. Box 368, Ridgefield, CT, USA), demonstrated efficacy in a phase I study of seven patients with moderate generalized PP (GPP), with all patients treated showing rapid skin improvement within 4 weeks after administration of a single dose [110]. In Effiyasil 1—a phase 2 trial—53 patients with GPP were randomized to receive either spesolimab (n = 35) or placebo (n = 18). At the end of week 1, 54% in the spesolimab group had a pustulation subscore of 0, as compared with 6% in the placebo group (*p* < 0.001). Infections occurred in 17% of the patients receiving spesolimab through the first week and in 47% at week 12 [111]. A phase IIa, multicenter, double-blind, randomized, placebo-controlled pilot study assessed the efficacy of different intravenous doses of spesolimab in 12 patients with PPP. At week 12, the primary endpoint was not met, although there was a faster decline in severity in the spesolimab groups [112]. Additional phase II and III studies of spesolimab are currently being performed in patients with PPP, GPP, HS, ulcerative colitis, and Crohn’s disease [113,114,115,116,117,118,119,120,121].

Results from a phase II study to evaluate the efficacy and safety of Imsidolimab (ANB019, AnaptysBio, 10770 Wateridge Circle Suite 210, San Diego, CA, USA) in PPP patients (POPLAR) have been recently posted; no severe adverse events were identified, but the primary outcome (change from baseline in PPP psoriasis area severity index) was not met [122,123]. Nevertheless, results from a phase II in six patients with GPP treated with imsidolimab seem more interesting [124]. Currently, there are ongoing phase II and III studies in HS and GPP [125,126,127].

Blockade of the IL-36 pathway using other mechanisms has been assessed in preclinical studies of several drugs such as short peptide IL-1R inhibitors [128] and small molecule blockers of IL-36y [129].

Potentiation of IL-38 by administering recombinant full-length IL-38 has shown anti-inflammatory action in murine models [39].

Finally, blockade of IL-1RAcP (also called IL-1R3) with a humanized monoclonal antibody (MAB-hR3) has been investigated, demonstrating reduction of proinflammatory cytokines both in vitro and in vivo [20]. Treatment with a chimeric mouse monoclonal antibody (MAB-mR3) in murine models of inflammatory diseases driven by IL-1β, IL-33, and IL-36 showed significant disease improvement [20].

## 5. Conclusions

The IL-1 family is a complex group of cytokines, receptors, and co-receptors with pro-inflammatory and anti-inflammatory properties, especially involved in maintaining endogenous homeostasis, activating the innate immune system, and providing links to switching on the adaptive response. Disruption of this equilibrium is involved in the pathogenesis of several inflammatory skin diseases, some of which have been reviewed herein.

Increasing knowledge of the IL-1 family cytokines and their regulatory mechanisms has allowed to develop different therapeutic strategies. Current and future trials will determine if these targets are useful in several inflammatory dermatoses, but preliminary results in psoriasis, AD, and HS are interesting. Targeting of IL-1 cytokines has been long studied, with mixed results of isolated experiences in GPP and HS, but the results of the APRICOT trial with anakinra in PPP have been disappointing. A subanalysis on patients with plaque psoriasis could provide more answers on the usefulness of IL-1 blockade in psoriasis [130]. The focus is now on anti-IL36R inhibitors, with an assortment of trials in PPP, GPP, HS, ulcerative colitis, and Crohn disease ongoing. IL-36R blockade has provided good results in phase I and II studies in GPP, but the primary endpoints have not been met in phase II studies on PPP [112,122]. The potential efficacy of IL-18 blockade in adult-onset Still disease and of IL-33 blockade in AD are being assessed in clinical trials. Finally, novel therapeutic approaches are underway, and future studies will provide more information on therapeutic targeting of IL-38 and IL-1RAcP.

## Figures and Tables

**Figure 1 ijms-23-09479-f001:**
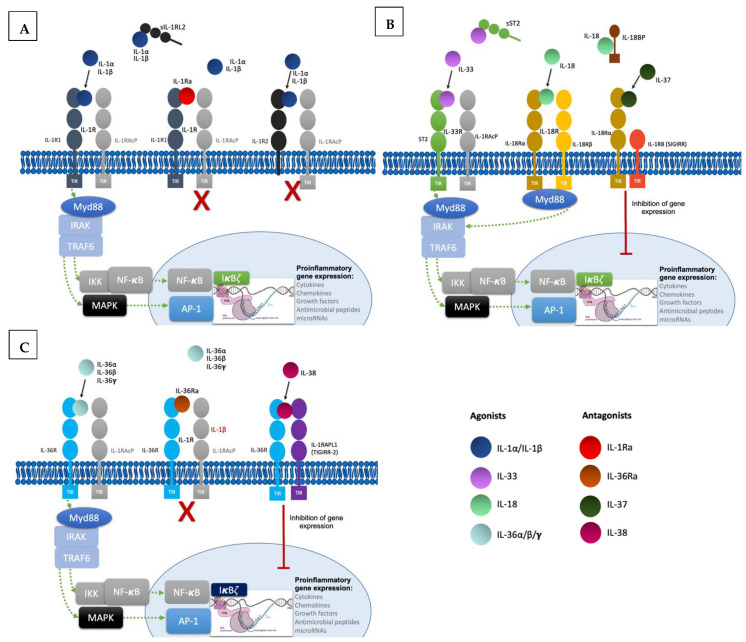
**Signaling pathways and regulatory mechanisms involved in the IL-1 family**. (**A**). Upon binding of IL-1α and IL-β to their receptor IL-1R1 altogether with co-receptor IL-1RAcP, induction of signal transduction with recruitment of myeloid differentiation primary response 88 (MyD88) accessory protein and IL-1R associated kinase (IRAK) proteins, ultimately ending in the activation of the transcription factor nuclear factor κB (NF- κB) and the transcription of proinflammatory genes. Regulatory mechanisms include IL-1R2 and IL-1Ra: IL-1R2 can exist as a soluble receptor or membrane bound, acting as a decoy receptor as it is unable to recruit the co-receptor to induce signal transduction. Finally, IL-1Ra acts as a competitive inhibitor by binding to IL-1R1. (**B**). Likewise, IL-33 binds to the receptor ST2, inducing the recruitment of co-receptor IL-1RAcP and resulting in signal transduction into the nucleus with transcription of proinflammatory genes. In this family, the soluble form of ST2 also acts as a decoy receptor. IL-18 binds to IL-18Rα and recruits the co-receptor IL-18Rβ resulting in pro-inflammatory signaling. IL-18 binds to the soluble protein IL-18BP preventing binding to the receptor. IL-37 is an anti-inflammatory cytokine and upon binding to IL-18Rα induces recruitment of the Single Ig and TIR Domain Containing (SIGIRR or IL-1R8), ultimately producing inhibitory signaling. (**C**). IL-36 cytokines also induce pro-inflammatory gene transcription by binding to the receptor IL-36R and recruiting co-receptor IL-1RAcP. IL-36Ra is the competitive antagonist of IL-36 cytokines. The anti-inflammatory cytokine IL-38 forms a complex with IL-36R and three immunoglobulin domain-containing IL-1 receptor-related 2 (TIGIRR-2 or IL1RAPL1), also inducing inhibitory signaling to regulate the pro-inflammatory gene activation.

**Table 1 ijms-23-09479-t001:** **IL-1 family cytokine members**.

Cytokines		Receptors (Other Names)	Co-Receptors (Other Names)	Function
IL-1	IL-1α	IL-1R1	IL-1RAcP (IL1-R3)	Pro-inflammatory
	IL-1β	IL-1R2		
	IL-1Ra	IL-1R1	N/A	Antagonist
IL-18		IL-18Rα (IL1-R5)	IL-18Rβ (IL-18RAcP or IL1R7)	Pro-inflammatory
IL-33		ST2 (IL-33R or IL1-R4)	IL-1RAcP	Pro-inflammatory Th2 responses
IL-36	IL-36α	IL-36R (IL-1Rrp2 or IL-R6)	IL-1RAcP (IL1-R3)	Pro-inflammatory
	IL-36β			
	IL-36γ			
	IL-36Ra	IL-36R (IL-1Rrp2 or IL-R6)	N/A	Antagonist
IL-37		IL-18Rα (IL1-R5)	IL-1R8 (SIGIRR or TIR8)	Antagonist
IL-38		IL-36R (IL-1Rrp2 or IL1-R6) IL-R9	IL1RAPL1 (TIGIRR-2) IL1RAPL2 (TIGIRR-1)	Antagonist/anti-inflammatory

SIGIRR: single immunoglobulin IL-1R-related molecule; TIR: toll-IL1R; IL1RAPL1 and IL1RAPL2: IL-1 receptor accessory protein like 1 and 2; TIGIRR 1 and 2: three immunoglobulin domain-containing IL-1 receptor-related 1 and 2.

**Table 2 ijms-23-09479-t002:** **Ongoing clinical trials of anti-IL36R inhibitors**.

Drug (Mechanism of Action)	Disease	Clinical Trial Number
Spesolimab (Effiyasil™) (BI655130) Humanized monoclonal antibody targeting IL-36R	GPP	NCT02978690
	PPP	NCT03100903
	PPP	NCT03135548
	Ulcerative colitis	NCT03482635
	GPP	NCT03782792
	PPP	NCT04015518
	Crohn’s disease	NCT04362254
	GPP	NCT04399837
	PPP	NCT04493424
	HS	NCT04762277
	HS	NCT04876391
	Crohn’s disease	NCT05013385
	GPP	NCT05200247
	GPP	NCT05239039
Imsidolimab (ANB019) Humanized monoclonal antibody targeting IL-36R	PPP	NCT03633396
	Acne vulgaris	NCT04856917
	HS	NCT04856930
	GPP	NCT05352893
	GPP	NCT05366855

GPP: generalized pustular psoriasis; PPP: palmoplantar pustulosis; HS: hidradenitis suppurativa.

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
