# Peer review of "IL-1 Family Cytokines in Inflammatory Dermatoses: Pathogenetic Role and Potential Therapeutic Implications"

_ijms, 2022, doi:10.3390/ijms23169479_

Round 1

Reviewer 1 Report

The paper itself is well written and provides sufficient description to enhance understanding in this area. Although the figures are color-coded for each cytokine and its receptor, they are not easy to understand at a glance. I would like to see an explanation of which cytokine is indicated by each color in a separate box in the figure.

Author Response

We would like to express our thanks to the reviewers for their feedback on our manuscript, and we are grateful to have been given the opportunity to respond and resubmit a revised version based on this feedback.

Please find, itemized below, our responses to each of the reviewers’ comments. All revisions to the manuscript have been made using tracked changes and have been approved by all co-authors.

Reviewer 1

The paper itself is well written and provides sufficient description to enhance understanding in this area. Although the figures are color-coded for each cytokine and its receptor, they are not easy to understand at a glance. I would like to see an explanation of which cytokine is indicated by each color in a separate box in the figure.

Thanks for this suggestion. We have added a color-coded legend box next to the figure.

Reviewer 2 Report

The authors present a clear and well-written review regarding the IL-1 family of cytokines, and their role in inflammatory skin diseases. The message of the various sections is clear, and there are only minor errors that require attention before the article may be published. These are:

1. The legend to the first table is shifted, so that there is part of it that is above the table. This should be corrected. In line 45 there is a comment regarding Figure 2 and Table 2 that should be removed, especially as there is no Figure 2. 

2. The Figure legend for Figure 1 appears to be the same format as the body text, this should probably be in a different format as per the instructions of the journal. 

3. The second sentence of section 2.2 (lines 97-98) would be better at the end of this paragraph. 

4. Abbreviations are at times introduced more than once, please check the manuscript so that once an abbreviation is introduced it is then used in that form. 

Notwithstanding these minor points the manuscript is well written and should be accepted once the abovementioned points are addressed. 

Author Response

We would like to express our thanks to the reviewers for their feedback on our manuscript, and we are grateful to have been given the opportunity to respond and resubmit a revised version based on this feedback.

Please find, itemized below, our responses to each of the reviewers’ comments. All revisions to the manuscript have been made using tracked changes and have been approved by all co-authors.

Reviewer 1

The authors present a clear and well-written review regarding the IL-1 family of cytokines, and their role in inflammatory skin diseases. The message of the various sections is clear, and there are only minor errors that require attention before the article may be published. These are:

  1. The legend to the first table is shifted, so that there is part of it that is above the table. This should be corrected. In line 45 there is a comment regarding Figure 2 and Table 2 that should be removed, especially as there is no Figure 2. 

These errors have been corrected. Thanks for noticing and letting us know.

  1. The Figure legend for Figure 1 appears to be the same format as the body text, this should probably be in a different format as per the instructions of the journal. 

Format has been corrected.

  1. The second sentence of section 2.2 (lines 97-98) would be better at the end of this paragraph. 

The sentence has been moved to the suggested paragraph.

  1. Abbreviations are at times introduced more than once, please check the manuscript so that once an abbreviation is introduced it is then used in that form. 

Abbreviations have been checked throughout all the manuscript and corrected.

Reviewer 3 Report

Dear authors,

This review summarizes the role of IL-1 family in different dermatological diseases. The manuscript is well written and structured and provides recent information about the immunological regulation of IL-1 in these diseases. However, the format of manuscript must be reviewed. The text must be justified, and many formatting errors must be checked. For example, lines 45, 130, among others.

In addition, I recommend to the authors to add a list of abbreviations to facilitate the reading of the manuscript. I also recommend that the foot of figure 1 be reviewed and each of the abbreviations used in it explained.

Author Response

We would like to express our thanks to the reviewers for their feedback on our manuscript, and we are grateful to have been given the opportunity to respond and resubmit a revised version based on this feedback.

Please find, itemized below, our responses to each of the reviewers’ comments. All revisions to the manuscript have been made using tracked changes and have been approved by all co-authors.

Reviewer 3: This review summarizes the role of IL-1 family in different dermatological diseases. The manuscript is well written and structured and provides recent information about the immunological regulation of IL-1 in these diseases. However, the format of manuscript must be reviewed. The text must be justified, and many formatting errors must be checked. For example, lines 45, 130, among others.

            The format has been checked and corrected.

In addition, I recommend to the authors to add a list of abbreviations to facilitate the reading of the manuscript.

            A list of abbreviations has been added to the manuscript.

I also recommend that the foot of figure 1 be reviewed and each of the abbreviations used in it explained.

            The foot of figure 1 has been reviewed and an explanation of each abbreviation has been explained.
